# Growth Hormone and the Auditory Pathway: Neuromodulation and Neuroregeneration

**DOI:** 10.3390/ijms22062829

**Published:** 2021-03-11

**Authors:** Joaquín Guerra Gómez, Jesús Devesa

**Affiliations:** 1Otolaryngology, Medical Center Foltra, 15886 Teo, Spain; 2Scientific Direction, Medical Center Foltra, 15886 Teo, Spain

**Keywords:** growth hormone, IGF-I, central auditory processing, hearing impairment, hereditary hearing loss, GH deficiency

## Abstract

Growth hormone (GH) plays an important role in auditory development during the embryonic stage. Exogenous agents such as sound, noise, drugs or trauma, can induce the release of this hormone to perform a protective function and stimulate other mediators that protect the auditory pathway. In addition, GH deficiency conditions hearing loss or central auditory processing disorders. There are promising animal studies that reflect a possible regenerative role when exogenous GH is used in hearing impairments, demonstrated in in vivo and in vitro studies, and also, even a few studies show beneficial effects in humans presented and substantiated in the main text, although they should not exaggerate the main conclusions.

## 1. Introduction

Hearing begins when the hair cells (HC) located in the cochlea perceive the sound stimulus. This sensory organ transduces the mechanical energy received for its transformation into electrical energy. The auditory pathway is responsible for the transmission of these acoustic signals to the brain, where the information received is processed for conscious perception. This path (afferent) has an opposite path (efferent) from which the received signal is regulated or inhibited. Both routes experience multiple interhemispheric crossings during their trajectory [1]. The physiological process of sound transmission involves the participation of neurotransmitters, in particular glutamate, but also acetylcholine or enkephalins [2] The same occurs with the development and protection of the auditory pathway, processes in which multiple neurotransmitters, hormones, and peptides are also involved, some of which are expressed in the ear itself, as appears to be the case with growth hormone (GH) and its receptor (GHR), at least in chick embryos [3]. This is not surprising, since GH is a pleiotropic hormone expressed at the pituitary level and in a number of cells and tissues where it performs many different functions, far beyond those classically described [4]. Indeed, GH and GHR are widely expressed in the brain, where the hormone and its main mediator Insulin-like Growth Factor-I (IGF-I) play different key roles in fetal and postnatal brain development, maturation, and function [5,6,7]. Furthermore, GH expression has been detected in neural stem cells [8,9], in which the hormone induces the expression of IGF-I [9].

In addition to the effects of GH/IGF-I on normal brain development and functioning, this system is also involved in the repair processes of the brain after injury, as many studies have already shown good results after GH administration to experimental animals [10,11,12,13,14,15,16,17] and humans [18,19,20,21,22,23,24,25,26,27], even when there is no GH deficiency.

Given the significant effects of GH on the brain, it seems logical that this hormone also plays a significant role in the auditory pathway. Even and given that GH secretion experiences a continuous decrease from approximately 20 years to being practically undetectable around 60 years of age [4], it is feasible to assume that many of the degenerative processes associated with aging may be related to the deficiency of this hormone; among them could be the hearing loss that is often seen in old age [28,29].

Physiologically, the loss of hair cells (HC) begins in the early stages of life, and they can be clinically observed from the fifth decade of life [28,29]. There are also exogenous factors such as noise, drug toxicity, ischemia or direct trauma that can damage these HC in a reversible or irreversible way, accelerating the process of auditory degeneration. The ability of hair cells to regenerate spontaneously seems not to be possible or really poor in mammals. However, this regeneration has been seen in other animal species, such as birds, reptiles, amphibians and fish [30], after exposure to noise or another ototoxic agent.

In this study we will analyze the role of GH in hearing and the physiological pathways that it regulates, as well as some syndromes related to GH that directly or indirectly condition hearing loss. Finally, we will assess the therapeutic potential of GH in the treatment of auditory pathology.

## 2. Physiological Role of GH in the Cochlea and the Auditory Pathway

Although it had been postulated that GH could be a neurohormone given its cerebral expression [3], most of the GH that exerts its actions at the brain level comes from that secreted by the pituitary gland. Plasma GH easily crosses the blood-brain barrier (BBB) to carry out its actions on neurogenesis, synaptogenesis and myelination, as well as to try to repair brain damage. In the case of IGF-I, although it is also expressed in the brain, its expression depends mainly on GH [4]. As with GH, plasma IGF-I is able to cross the BBB, but only the free fraction of the circulating hormone can do so, therefore its brain concentrations depend on the molar ratio IGF-I/IGFBP3 (its main carrier in plasma). Furthermore, GH induces the expression of several neurotrophic factors that are involved in various regulatory functions of the auditory pathway [31].

Studies in animal models have shown the relevance of GH in the development of the organs of the inner ear. During the embryonic period, the chicks express high GH immune reactivity in the vesicles that will develop the inner ear [3]. During postnatal development in mice, transcriptomic analysis shows an increased level of GH (89) in the cochlear sensory epithelium [32]. Furthermore, during growth and aging, cochlear upregulation of GH (44) occurs along with upregulation of prolactin (108) [33]. GH is a hormone that regulates calcium mineralization and it is also involved in the growth of the temporal bone and the ossicular chain of the middle ear [34]. It is also responsible for the mineralization of otoconia, a calcium carbonate structure located in the vestibular system that informs the cephalic positions, through matrix-related sequences [35]. With respect to the central auditory effects of GH, peripheral administration of its secretagogue GHRH has shown a reduction in late auditory evoked potential latencies in young healthy men, concomitant with an increased circulating GH levels [36].

## 3. GH and Hearing Impairment

Interestingly, hearing is directly related to the growth rate in children and adolescents. In fact, the hearing threshold in infancy can predict growth rate [37]. There are several syndromes with hearing dysfunction in which there is GH deficiency, however this is not a constant for syndromes in which hearing loss exists [38]. That is, depending on the syndrome, there could be an absence or insufficient GH, or a resistance or decreased sensitivity to the hormone. In these cases, there is a deficit of IGF-I, and hearing impairment is mainly related to the decrease in the levels of this GH mediator [37,38,39].

Adults with untreated, congenital lifetime isolated GH deficiency usually show predominance of mild high-tones sensorineural hearing loss [40]. Neurophysiologic tests show signs of cochlear injury, such as the absence of the stapedial reflex and otoacoustic emissions, also seen in Laron syndrome [41]. The absence of the stapedial reflex implies hypersensitivity to noise (hyperacusis) [40,41,42]. Some syndromic children with impaired GH secretion may have a temporary conductive hearing loss [43].

Both GH deficiency and excess involve an alteration in central auditory processing, leading to cognitive deficits. GH-deficient subjects show an increase of the acoustic P300 latency, whereas in those with acromegaly there is a reduction in the amplitude of this wave, which also conditions cognitive impairment [44]. Brain trauma can also cause a deficit in GH synthesis and secretion. After suffering a head injury, some boxers and kickboxers become GH deficient. In them, the P300 wave amplitudes are lower than in those without GH deficiency [45].

All of the evidence described above demonstrates that GH plays a major role in the entire auditory pathway, corroborating the findings about untreated GH deficient children and hearing loss [42].

## 4. The Sound as a Modulator of GH Secretion

The fact that noise damages the auditory pathway is undoubted, but the mechanisms by which this occurs are not yet well known. The effects of noise on the auditory pathway usually appear rapidly, damaging synapses, inducing the appearance of reactive oxygen species (ROS), calcium accumulation, and recruiting inflammatory mediators (such as IL-6 or TNF) and increasing glutamate excitotoxicity. Noise activates several pathways including caspase or JNK/MAPK dependent and independent pathways [46,47,48]. Noise also induces cognitive dysfunction that results in the late onset of hippocampal neurogenesis, which develops long after oxidative stress disappeared [49]. This region is more susceptible to noise than the auditory cortex, with greater damage from oxidative stress and oligomerization and hyperphosphorylation of Tau, a microtubule-associated protein [50], which can lead to progressive synaptic loss and neuronal cell death [51,52]. This pathological accumulation of Tau may be correlated with neurodegeneration in the peripheral auditory system [53].

A series of studies showed that prolonged exposure to high intensity noise initially upregulates GH expression [54], increasing its circulating levels, as well as other stress-related hormones, both in humans and in animal models [55,56]. Interestingly, a short but intermittent exposure to a low frequency, moderate intensity sound stimulus increases nerve motility and cellular response, implicating higher circulating GH levels [57]. However, in the long term, these levels are significantly reduced [58]. The decrease in circulating GH is critical in individuals who are growing and maturing, potentially altering their neuronal plasticity and, therefore, affecting their learning and memory [59].

Based on these results, it seems that after exposure to noise, GH can play an important role in neuroprotection and neuroregeneration, at least in animal models, so the next step is to consider whether exogenous GH could exert viable clinical effects to regenerate the auditory pathway.

## 5. The Effects of GH: From In Vitro/In Vivo Studies to Clinical Reports

In vivo and in vitro studies have tried to analyze whether GH could protect, recover or even regenerate cells involved in hearing. The first studies focused on observing the ability of GH to successfully stimulate the proliferation of cultured outer ear cartilaginous structures (chondrocytes) of rabbits, showing positive results suggesting that GH directly initiates proliferation in rabbit ear chondrocytes and, consequently, in mammals [60].

However, few reports have demonstrated the efficacy of exogenous GH in hearing. In fact, one might even wonder if treatment with this hormone may pose a risk of hearing impairment since, theoretically, the increase in GH induces greater ossification of the ear cavities that could worsen auditory transmission, as apparently seen in acromegaly [61]. However, GH treatment does not increase the risk of hearing loss, either in infants or children or in adults, as has been seen in patients with Turner syndrome and other conditions [62,63] (Table 1). In fact, the incidence of middle ear pathology often decreases with accelerated bone mineralization of the ear cavities in these girls with Turner syndrome [34].

A brain injury, such as traumatic brain injury or cerebral palsy, can lead to a deficit of GH, which implies a replacement treatment with this hormone that, as previously mentioned, is safe and effective in brain repair. In fact, as our group demonstrated, GH treatment is able to restore hearing in patients with cerebral palsy and hearing loss [64]. Its mechanism of action is unknown, but it probably occurs through the regeneration of sensory cochlear cells and the auditory nerve or by accelerating hearing maturity [64]. Furthermore, GH induces the expression of IGF-I, and other neurotrophic factors, which play a relevant role in hearing; circulating levels of IGF-I are directly proportional to the degree of hearing loss in older humans [65]. Moreover, topical administration of IGF-I had been seen to have positive effects on hearing recovery in patients with sudden sensorineural hearing loss [66].

The therapeutic potential of GH for use in other conditions such as ischemia, noise-induced hearing damage, or drug toxicity needs to be evaluated but appears promising. Early treatment is undoubtedly a factor to ensure its success [64].

### 5.1. Hair Cells

GH administration can stimulate the proliferation of HC after cochlear damage. Zebrafish has been the most studied species of all animal models due to the inherent ability of this fish to induce neuroregeneration. In zebrafish, the use of GH antagonists leads to a significant decrease in cell proliferation in the inner ear. Faced with a stimulus that generates cochlear damage, and damage to hair cells, endogenous GH can increase its expression, to stimulate the regeneration of those auditory hair cells. GH mRNA expression in this case is located perinuclearly around erythrocytes in the blood vessels of the inner ear epithelium. After overexposure to noise, the densities of the saccular hair cells in the GH-treated treated fish were similar to those of the controls, and higher than those of the fish that received buffer. With the use of exogenous GH, cell proliferation was stimulated and apoptosis was reduced in the saccules, lagenae and utricles of the fish treated from the day after acoustic trauma [54,67].

### 5.2. Auditory Nerve

Another study has focused on evaluating the effects on nerve regeneration of neurons of the spiral ganglion, a group of bipolar cell bodies that innervate the ciliated cells of the organ of Corti and project axons to the cochlear nerve. This in vitro study showed that exposure to different concentrations of GH induces branching and cell growth. Regarding an effect of GH on neuronal survival, while it increased after the combined administration of GH and BDNF (Brain Derived Nerve Factor), the administration of only GH failed to reproduce this effect [68].

### 5.3. Deep Brain

The hippocampus is an area of special interest in the auditory pathway since neural circuits connect to this brain structure. Neurons in the hippocampus respond to to action potentials generated in the inner ear following sound stimuli and are not tonotopically organized [69]. During auditory working memory (including coding, maintenance, and retrieval processes), the auditory cortex connects with the hippocampus and the inferior frontal gyrus [70]. This auditory information travels from the auditory cortex to neurons in the median septum and is then projected directly to the hippocampus [69]. Interestingly, studies in rats show that GH is capable of reversing hippocampal dysfunction due to various stress-related conditions, including long-term auditory fear memory [71]. This observation about the effect of GH on excessive fear memory formation has also been reported in the brain amygdala [72].

### 5.4. Auditory Cortex

There is a promising prospect for the recovery of auditory processing in GH-deficient patients through the use of GH replacement therapy. Such is the case in patients with Sheehan syndrome. These patients have panhypopituitarism resulting from an infarction of the pituitary gland due to hypovolemic shock or severe hemorrhage. Neurophysiological studies show that the administration of GH decreases the prolongation of P300 latencies after six months of treatment [73]. GH treatment may also improve the frequency of speech in Turner syndrome [74].

## 6. Molecular Mechanisms of GH-Dependent Neuroprotection, Synaptogenesis and Neurogenesis

As mentioned in the Introduction, it is well established that GH gene expression occurs in many tissues and organs, including the central and peripheral nervous system, and this expression is independent of that which occurs in the pituitary gland. GH administration is capable of increasing axonal regeneration, promoting reinnervation, maintaining or inducing Schwann cells proliferation, and reducing muscle atrophy; all this translates into great potential for the treatment of peripheral nerve injuries [75,76,77,78]. Since the basis of auditory regeneration seeks molecular targets that allow the development of drugs to restore lost hearing, GH could be one of them.

The neuroprotective and neuroregenerative effects of GH are carried out: 1) by the direct action of this hormone, or 2) by inducing the expression of neurotrophic factors. The direct action of this hormone begins after the binding of GH to its receptor, which leads to the activation of the JAK/STAT signaling pathway, considered as the main intracellular transduction system of the effects of GH, although also the signaling pathways PI3K/Akt, MAPK and Notch are activated by the hormone and play an important role in the effects of this hormone [79,80,81]. Phosphorylated STATs form dimers that enter the nucleus, where they bind to specific DNA sequences and activate their target genes and other signaling pathways [80]. This activation mechanism has been demonstrated in zebrafish, promoting regeneration of hair cells [82]. In addition, GH has shown to exert anti-inflammatory actions, for example by modifying the immunoreactivity of TNF receptors [83], as well as modifying neuronal excitation and neuroplasticity through up-regulation of the NR2B subunit of the NMDA receptor [71], and it also increases the number of neurons that increase the expression of c-Fos in BLA (basolateral complex amygdala) [72].

As indicated, GH, via STAT5, activates the Notch signaling pathway, which has been shown to be useful in retinal neuroprotection [83]. This effect is clearly demonstrated by the fact that, cotreatment of GH with the Notch signal inhibitor, DAPT, inhibits the neuroprotective effect of the hormone [83]. In the same study, the authors found another neuroprotective effect of GH, exerted via Notch, by the phosphorylation of the PI3K/Akt signaling pathway that reduces the regulation of PTEN, thus decreasing the inflammatory response to kainic acid administration. This study has been carried out in the retina of chicken in which GH also promotes retinal regeneration after damage induced by kainic acid [83]. Therefore, it does not seem that this could be extrapolated to what happens in the human inner ear, although the retina and inner ear are very specific sensory structures. In fact, in mice, Notch inhibits Atoh1, a relevant factor that plays a major role in the development and regeneration of hair cells [84], however the same study indicates that additional factors are needed for Atoh1 to exert its regenerative effects at the level of certain genes present in HC. Most likely, the type of relationships between Notch and Atoh1 has led to investigate on how to inhibit the target gene *Hes5* or γ-secretase to significantly increase the expression of Atoh1 in the cochlea [85,86].

As indicated above, GH can induce the release of other neurotrophic factors, namely IGF-I and neuropeptides, hormones, neurotrophins, growth factors, and some cytokine signaling suppressors [80]. IGF-I is one of the main regulatory mediators of hearing function. This factor, along with its receptor (IGF-IR), are expressed during cochlear development. IGF-I signaling ensures growth and differentiation of cochlear structures. Although the lack of significant amounts of IGF-I alters gene expression in hair cells, partial signaling is sufficient for their normal development [87]. But IGF-I also plays a relevant role in the myelination of the auditory nerve, activating the PI3K/Akt pathway. This promotes differentiation through the expression of myelin core protein and myelin-associated glycoprotein [88]. Therefore, it is likely that some or most of the effects of GH in the inner ear are dependent on GH-induced expression of IGF-I.

It is interesting to note that the neurotrophic effects of GH are also mediated by the induction of classical neurotrophins such as BDNF and NT3 [12,89], supporting the participation of GH in the recovery of functional synaptic transmission after damage. BDNF and NT-3 are expressed, together with their receptors tyrosine kinase B (TrkB) and tyrosine kinase C (TrkC), respectively, during the embryonic development of the inner ear, and participate in the sensory trophic support of this structure, providing innervation, development and maintenance [90,91]. Erythropoietin (EPO), a hormone also induced by GH, is also expressed in the inner ear, where in vitro studies in cultures of the organ of Corti of rats have shown that it protects hair cells from ischemia [92] or ototoxicity [93], although it lacked positive effects, and even worsened hearing loss, in in vivo studies in these animals after noise-induced damage [94]. Therefore, the effects of EPO at the level of the inner ear still need to be studied in more detail.

Although not directly related to GH, Heregulin (HRG), a member of the epidermal growth factor (EGF) family, has been reported to act as a mitogen. It stimulates the proliferation of utricular macules in mouse cultures, at least during the development of the neonatal inner ear. Its receptors are expressed in the inner ear of newborns and adults in vestibular end organs and Organ of Corti. However, the addition of HRG does not enhace cell proliferation in adults. [95].

## 7. Conclusions

Throughout this review, we see that listening is a complex process that involves multiple factors that act directly or indirectly. It is clear that GH participates in auditory embryonic development and this hormone is released by a sound stimulus or a harmful event. It also plays a role in central auditory processing. Along the same lines, the absence of endogenous GH can lead to hearing loss, while its replacement can help to recover hearing loss, as it has been seen in untreated GH-deficient children. Many children in the world, including Spain, do not receive GH replacement therapy despite that they are GH-deficient. The reasons for it are diverse: Health agencies or GH Committees require two negative GH provocative tests (these are few reliable), a pituitary MRI, and a specific age. Hence, many children suffer GH-deficiency and they are not treated because they do not meet the specified requirements. This is not logical but it is the real thing. We know a number of cases.

We know that multiple etiopathogenic factors cause hearing loss, including genetic and environmental disorders, which are not fully understood. Agents that promote protection from a traumatic event or degenerative process, or ideally regenerate the damage caused, have been investigated for several decades. The apparent irreversibility of hearing loss once the lesion is established forces us to search for molecules that slow down or recover the sensory cells and auditory neurons. Although studies in animal models are recent, their results and those in some GH-deficient children promise to be feasible for treatment with GH, alone or in combination with other factors. In fact, although GH might not have a direct effect, the hormone induces the expression of several factors, mainly IGF-I, which have positive actions on the recovery from hearing loss. In any case, more studies are needed to analyze in detail the effects that GH can have on a damaged cochlea.

## Figures and Tables

**Table 1 ijms-22-02829-t001:** Selection of hereditary hearing loss syndromes and the involvement of GH on their pathogenesis, associated conditions, or treatment.

Condition	Findings	Hearing Loss	GH Levels	Benefit with rhGH
CHARGE syndrome	Colobomatous microphthalmia Congenital heart defects, usually conotruncal Choanal atresia Retarded growth and development Genital hypoplasia, possibly of hypothalamic origin	SNHL MHL (rare)	Normal Low (rare)	Yes
Alström syndrome	Atypical retinal degeneration with loss of central vision in infancy Diabetes mellitus in childhood Transient obesity Posterior cortical cataract Nephropathy Acanthosis nigricans	SNHL	Low	Yes
Acrodysostosis	Shortening of the interphalangeal joints of the hands and feet Intellectual disability Peculiar facies (short head, small broad upturned nose with flat nasal bridge and protruding jaw) Increased bone age Intrauterine growth retardation Juvenile arthritis Short stature	SNHL (rare) MHL (rare)	Normal	Yes
Combined growth hormone deficiency with hearing loss and limited neck movement	Pituitary hormone deficiencyLimited neck movement	SNHL	Low	Yes
Laron syndrome	Insensitivity to GH, usually caused by a mutant growth hormone receptor Short stature Increased sensitivity to insulin	SNHL MHL (rare)	Normal	No
Intrauterine and postnatal growth failure with microcephaly and intellectual disability	Prenatal Growth Failure Elevated Growth Hormone Levels Mental Retardation	SNHL	Elevated	No
Hajduk-Cheney syndrome	Dissolution of terminal phalanges Dolichocephaly with occipital prominence Short stature Premature loss of teeth	CHL SNHL	Normal Low (sporadic)	Yes
Richards-Rundle syndrome	Ataxia Distal amyotrophy Intellectual disability Diabetes mellitus Absent development of secondary sex characteristics	SNHL	Normal	No
EEC syndrome	Variable ectrodactyly of hands and feet Absence of lacrimal puncta Cleft lip-palate Occasional vestibular abnormalities	CHL	Normal Low (sporadic)	Yes
Crandall syndrome	Generalized alopecia with pili torti Growth retardation Hypogonadism	SNHL	Low	Yes
Hypodontia and PEG-shaped teeth, olivopontocerebellar dysplasia, hypogonadism and hearing loss	Olivopontocerebellar degeneration Hypogonadotropic hypogonadism Hypodontia Resistance to GH	SNHL (unilateral)	Normal	No
Turner syndrome	Short stature Thick or webbed neck Gonadal dysfunction	CHL MHL SNHL	Normal	Yes
Kabuki syndrome	Peculiar facial (elongated palpebral fissures and prominent ears) Postnatal growth retardation Intellectual disability Cardiac anomalies	CHL MHL SNHL	Normal	Yes

SNHL: sensorineural hearing loss; CHL: conductive hearing loss; MHC: mixed hearing loss. Table adapted from Toriello, H et al. Hereditary hearing loss and its syndromes [38].

## Data Availability

The data presented in this study are available on request from the corresponding author.

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
