# Peer review of "Growth Hormone and the Auditory Pathway: Neuromodulation and Neuroregeneration"

_ijms, 2021, doi:10.3390/ijms22062829_

Round 1

Reviewer 1 Report

An excellent paper reviewing growth hormone in view of the auditory pathway.

Some suggested revisions based mainly on wording:

L40: explain abbreviation first time it occurs: IGF-1

L72: pituitary gland

You switch in the manuscript from IGF-1 to IGF-I. Please decide for one form.

L106: delete the second “adults”.

L114: show increase of the

Please move table 1 to a position below referring to it.  I was irritated where the reference in the text is when reading page 3 and 4.

L156: could exert, please delete “to”

L208 please rewrite the sentence. It is not correct that modiolus belongs to the OoC what due you mean by “shed the axons”? please rewrite.

L228: please delete one of the two “patients”

L 216: hippocampus does not respond to sound. Sound is transformed in the inner ear to action potential.

L301: ref 95 is a review. Please refer to the original studies: who reports on EPO and ischemia? Who on ototoxins and who on noise?

L319: what do you mean by “untreated” GH-deficient children? Why untreated?

Since the data on exogenous GH support to treat HL are rare you may include in the conclusion that there has to be more activity in future to elucidate the effects of GH therapy in traumatized cochlea.

Author Response

Thank you for your comments and suggestions. 

  • Grammatical mistakes have been corrected.
  • IGF-1 has been changed to IGF-I in the manuscript.
  • The sentence "shed the axons" has been corrected. 
  • Ref. 95 has been changed to three original reports and the explanations you asked for have been given. 
  • An explanation about untreated GH-deficient children has been given.
  • Table 1 has been moved to its correct position. 
  • Your suggestion for the Conclusions has been added to them.

Reviewer 2 Report

This is a review regarding the effects of growth hormone and its therapeutic implications for hereditary hearing syndromes in regard to neural protection and regeneration. The essay is well-organized and concise. However, there are a few suggestions that I would like to point out.
1.    From lines 302 to 310, the content is almost identical to sentences in the abstract of reference 96 (PMID: 14690060). I recommend the authors integrate materials in the literature and make conclusions in their own words.
2.    I suggest the authors to review the grammatical correctness and the coherence of the content. 
    For example,
   (i) In line 114, “increased” should be replaced by “an increase”, and the sentence could be more fluent and could convey messages more clearly after minor adjustments. 
    (ii) In line 183, “topic” should be replaced by “topical”. 
    (iii) In lines 210 and 211, the sentence “however, it was unsuccessful in neuronal survival”, could be clearer.  Similar problem in line 223, “This observation has also been reported in the brain amygdata”. What’s the observation ? Reversing the hippocampal dysfunction?  long-term auditory fear memory ?
    (iv) In line 305, there is a repeated word “not”. 
(v) In line 136, “which develops long after disappearing oxidative stress” — weird use of “disappearing” here. 
 (vi) In line 133, Noise “actives” several pathways..
(vii) In abstract, “There are promising research studies that reflect a possible regenerative role when exogenous GH is used in hearing impairments, demonstrated in in vivo and in vitro studies, and also, although scarce, positive reports in humans.”   The sentence is too long to read.  Besides, what do “positive” reports mean ?    Is “research studies” different from researches or studies ? 

3. Table 1 is a nice summary for GH and hereditary hearing loss syndromes. However, no reference was assigned. It’s better to have evidence or references for GH levels and benefit with rhGH.   

Author Response

Thank you for our comments and suggestions.   

  • Grammatical mistakes have been corrected.
  • The sentence "however, it was unsuccessful has been clarified, also it has been clarified the observation reported in the brain amygdala.  
  • The abstract has been corrected. - Reference of Table 1 has been included. 
  • Our conclusions concerning lines 302-310 has been adapted.
  • The effect of sound on hippocampus has been changed.